# Diagnostic Outcomes and Treatment Modalities in Patients with Mycosis Fungoides in West Sweden—A Retrospective Register-Based Study

**DOI:** 10.3390/cancers14194661

**Published:** 2022-09-25

**Authors:** Karolina Wojewoda, Martin Gillstedt, Hanna Englund, Shada Ali, Catharina Lewerin, Amra Osmancevic

**Affiliations:** 1Department of Dermatology and Venereology, Institute of Clinical Sciences, Sahlgrenska Academy, University of Gothenburg, 413 46 Gothenburg, Sweden; 2Department of Dermatology and Venereology, Sahlgrenska University Hospital, Region Västra Götaland, 413 46 Gothenburg, Sweden; 3Section of Hematology and Coagulation, Department of Internal Medicine, Institute of Medicine, Sahlgrenska Academy, University of Gothenburg, 413 46 Gothenburg, Sweden

**Keywords:** mycosis fungoides, cutaneous T-cell lymphomas, treatment, lymphomas, skin

## Abstract

**Simple Summary:**

Mycosis fungoides (MF) is a rare and incurable disease, but there are a wide variety of treatment options. Since the condition is rare, only a few studies have been performed on this topic in Sweden. This study contributes to the knowledge of the epidemiological and clinical features and diagnostic findings in addition to the treatment modalities and responses in patients with diagnosed MF and/or followed up for a long period of time in Sweden. The results of this study can be used to improve clinical practice and stimulate future research.

**Abstract:**

(1) Background: Mycosis fungoides (MF) is a variant of primary cutaneous T-cell lymphoma. The aim of this study was to describe the clinical features and epidemiological and diagnostic findings in addition to the treatment modalities and responses in patients with MF. Furthermore, comparisons between patients in the early stage and the advanced stage were evaluated. (2) Methods: A retrospective register-based study based on data collected from the primary cutaneous lymphoma register and medical records was performed at the Department of Dermatology and Venerology at Sahlgrenska University Hospital, Gothenburg, Sweden. (3) Results: Eighty-four patients with a median age of 55 years with MF were included. Most of the patients (*n* = 73) were diagnosed at the early stage of the disease (IA–IIA). Overall disease progression was seen in 12.5% (*n* = 9) of the patients. Nine (10.7%) patients were deceased, out of which four (4.8%) deaths were associated with MF-related causes. (4) Conclusions: This study contributes to the knowledge of the epidemiological and clinical features in addition to the diagnostic findings and treatment responses in patients with MF in Sweden.

## 1. Introduction

Primary cutaneous lymphomas (PCL) represent a heterogeneous group of extranodal non-Hodgkin lymphomas consisting of cutaneous T-cell lymphoma (CTCL) and cutaneous B-cell lymphoma (CBCL) that primarily affect the skin, with no extracutaneous involvement at the time of diagnosis [1]. The most common subtype of CTCLs is mycosis fungoides (MF) (55%).

CTCLs are a group of rare diseases with an incidence of 7.7 per million persons in the USA [2] and other countries showing similar numbers. The data by Korgavkar et al. from 2013 show that the incidence of CTCL has stabilized since 1998 [3]. There have been reports about an increased incidence of CTCL in the past decade. For example, studies performed in Norway during the period of 1980–2003, in Germany during the period of 2013–2014, and France during the period of 2005–2019 suggested that the incidences of MF and Sézary syndrome (SS) were increasing [4,5,6]. However, these studies were restricted to geographical populations [4,5] and subpopulations [6], limiting their validity.

MF is clinically characterized by the evolution of erythematous patches, plaques, and less frequently, tumors (Figure 1a–c). The lesions seem to have a predilection for areas that are not exposed to the sun, such as the buttocks, breasts, the inner part of the upper extremities, and the medial thighs, but any area of the skin can be affected [7]. The dermatoscopic features of MF are fine short linear vessels, orange-yellowish patches, and spermatozoa-like structures (Figure 1d) [8,9,10,11,12].

The median age at diagnosis is 55–60 years, and it is seen more frequently in males, with a male-to-female ratio of 1.6–2.0:1 [13]. The pathogenesis of MF remains only partially understood.

Diagnosing MF, especially early-stage MF, can be difficult, may require multiple types of diagnostic tools, and can take years. It can be very challenging for a pathologist to recognize MF when looking at a biopsy taken from the early patch stage. Usually, there is only a mild perivascular infiltrate in the upper dermis. When the patches mature and become thin plaques, epidermotropism (the movement of atypical lymphocytes into the epidermis) and lymphocytes are visible. The lymphocytes in the dermis may vary in size and shape, but no distinction can be made between tumor cells and normal cells in this stage. As the plaques grow visible, a subepidermal band of lymphocytes with cerebriform nuclei begins to appear, making them easier to detect under the microscope. Epidermotropism is now more prominent, and the intraepidermal accumulation of atypical lymphocytes, so called Pautrier microabscesses, can be seen in one-third of cases. In the final tumor stage, epidermotropism is lost, and the atypical cells are instead large and clustered in the dermis with an admixture of other lymphocytic cells [14,15,16].

For pathologists to be able to distinguish between the different types of lymphocytes they are seeing on a biopsy, they must make use of immunohistochemistry. The phenotype of the neoplastic cells in classic MF are CD2+, CD3+, CD4+, CD5+, CD8−, CD45RO+, CD20−, and CD30− [14,16,17,18].

The identification of T-cell receptor (TCR) clonality, in addition to histopathology and immunophenotype, is of great diagnostic value when working with MF. The clonality is determined by detecting alpha/beta or gamma/delta TCR gene rearrangements with the help of a PCR technique. The results reveal if the neoplastic cells arise from the same precursor cell. Previous studies established that the likelihood of finding clonality increases with disease progression and that clonality was found in 100% of samples in the tumor stage [19], 73% of samples in the infiltrated stage [19], and 52% to 75% of samples in the patch/plaque stage [20,21]. It is important to note that clonal T-cell populations may also be found in benign dermatoses, and therefore it is important to consider its correlation with histological and clinical findings [22].

There are several international guidelines on the diagnostics and treatments of PCL [23,24], and the Swedish Regional Cancer Centre (RCC) released national guidelines in 2019 [25]. Studies on the epidemiology, diagnostics, and clinical features of PCL in Swedish patients are scarce [26,27,28].

Patients with MF can be categorized into two groups based on the initial staging. Stages IA–IIA relate to a more indolent course, while stages IIB–IVB relate to a more progressive development [29]. The advanced stages are unfortunately associated with shorter overall survival [29]. MF remains an incurable disease, and treatment aims to ameliorate symptoms and improve disease-related quality of life. The current perception is that patients with early-stage disease should primarily be treated with skin-directed therapy (SDT). Systemic therapy should instead be administrated to those with advanced-stage disease or refractory disease. It is also possible to combine a systemic treatment with SDT. In terms of life expectancy, the prognosis is excellent for those with limited-stage disease [13].

Still, the care of these patients has been steered by local traditions and the practitioner’s experiences. It has been stated that there is a lack of descriptive data from Sweden concerning patients with MF. This study creates an overview of the treatment modalities used in patients with MF in West Sweden. The primary aim of this report was to describe and characterize the epidemiology, clinical features, histopathology, immunophenotypes, and molecular findings of MF diagnosed and/or followed up at the Department of Dermatology and Venerology at Sahlgrenska University Hospital between 2005 and 2022.

The secondary aim of this project was to describe the treatment modalities in patients with MF in West Sweden and evaluate the treatment responses.

## 2. Materials and Methods

This is a descriptive, retrospective, register-based study of patients diagnosed with MF. A total of 143 patients (including deceased patients) with an initial diagnosis of PCL were identified from the PCL register in West Sweden between 1 January 2005 and 1 July 2022 and had, at some point, been referred to the dermatologic clinic at Sahlgrenska University Hospital. The PCL register at the Department of Dermatology and Venerology, Sahlgrenska University Hospital, was founded in 2014. Into this register, patients with any PCL were continuously added. After a systematic review of the medical records and the results from the histological analysis of skin biopsies, 48 patients were excluded due to having a PCL diagnosis other than MF. Another 11 patients were ultimately excluded due to having uncertain histological and/or clinical findings, thus reducing the final study population to a total of 84 (Figure 2).

Basic characteristics, such as age, sex, the debut of symptoms, smoking habits, profession, and comorbidity, were collected and registered along with the clinical findings and the tumor–node–metastasis–blood (TNMB) stage at the time of diagnosis. The histological assessments of skin biopsies regarding epidermotropism and atypical lymphocytes were made by a pathologist. Up to four histopathological diagnosis slides were recorded. The immunohistochemistry was evaluated, and the performed staining focusing on the detection of cluster of differentiation (CD): CD2+, CD3+, CD4+, CD5+, CD7, CD8−, CD20−, and CD30−, among others. TCR clonality was evaluated as monoclonal or polyclonal. For the monoclonal results, the TCR gene rearrangement for gamma and beta were recorded as either present or not present. All treatments provided were registered along with the treatment responses.

The date of debut was set to the subjective estimation of when the symptoms of disease appeared for the first time. In the case of the absence of information or uncertainty about the onset of symptoms, the date was set to the middle of the month or year, e.g., 15 June 2020. Patients who had not previously been staged at the time of diagnosis were staged according to the current World Health Organization—European Organization for Research and Treatment of Cancer (WHO-EORTC) classification based on the available information in medical records and photodocumentation at the time [1,13]. No consideration was given to the evolution or eventual transition of the patient’s disease during the staging process. Study objects that had not been fully investigated with fluorescence-activated cell sorting or had not undergone the histological confirmation of suspicious metastases were assigned B0 or M0, respectively.

The patients were divided into two groups based on if they were staged with early-stage (IA–IIA) or advanced-stage (IIB–IVB) disease at the time of diagnosis. When describing the provided treatment, each patient was only counted once, even if the treatment had later been repeated in that specific patient.

The treatment response was evaluated from the response in the skin, based on the consensus statement of the International Society for Cutaneous Lymphomas (ISCL) and EORTC for response criteria in MF [30]. The treatment response was estimated from descriptions in medical records and follow-up photodocumentation. There was no fixed time interval for evaluating the treatment response. However, the evaluation was predominantly estimated from documentation from the follow-up visit. The time range for each patient’s follow-up visit generally varied between three and twelve months, depending on the severity of the disease. The treatment response in patients with combination therapy was evaluated from the time point when the observed treatment was first added to the patient’s ongoing therapy regimen.

All data were analyzed using R version 3.5.3 (The R Foundation for Statistical Computing, Vienna, Austria). Wilcoxon’s rank sum test was used for two-sample comparisons. Fisher’s exact test was used for comparing proportions. Kaplan–Meier plots were generated for overall survival, and a Cox proportional hazards regression was used to compare survival between early and advanced staging. All tests were two-sided, and *p* < 0.05 was considered to be statistically significant. Microsoft Excel was used to create the tables and graphs. Descriptive statistics, such as baseline characteristics (age and staging), are presented as medians and ranges, sex is presented as percentages, and 95% confidence intervals (CI) were computed.

## 3. Results

### 3.1. Patient Demographics

An overview of the demographic and clinical characteristics is presented in Table 1. Eighty-four patients with clinically and histologically verified MF were identified in West Sweden between 1 January 2005 and 27 June 2022. Hypopigmented MF was seen in one patient, folliculotropic MF was seen in nine (10.7%) patients, and syringotropic MF was seen in four (4.8%) patients. CD30-positive transformations were seen in seven patients (8.3%). The remaining 63 (75%) patients were assessed as classic MF. The median age at the time of diagnosis was 55 years (range 9–92 years). The male-to-female ratio was 1.8:1.

The most common comorbidities in our study population (*n* = 84) were cardiovascular diseases such as hypertension (32%), hyperlipidemia (19%), and heart failure (8%) and diabetes mellitus (14.3%). Moreover, ten (11.9%) patients suffered from depression. Twenty-five (29.8%) patients had other malignant comorbidities: skin cancers occurred 28 times, hematological malignancies occurred 9 times, and solid tumors occurred 13 times in patients with MF (Table 2). The most common skin cancer was basal cell carcinoma, which occurred in 14 (16.7%) different patients. Benign skin conditions other than MF occurred in 40 patients (47.6%).

According to the WHO criteria of weight for adults (underweight (less than 18.5 kg/m^2^), normal weight (18.5 to 24.9 kg/m^2^), overweight (25.0 to 29.9 kg/m^2^), and obesity (30 kg/m^2^ or greater)), most patients (41%) were overweight and 21% of patients were obese, whereas only 36% were in the normal body mass index (BMI) group (*n* = 73).

The majority of the patients (53%) had never smoked, 8% were current smokers, and 38% of patients were previous smokers (*n* = 60).

Data on previous or current professions or places of work were available for 70 patients. The majority were teachers (11.4%), mechanics or working with cars (11.4%), office workers (10%), students (8.6%), and other type of workers.

### 3.2. Clinical Characteristics at the Debut

Time from onset of skin symptoms until the initial diagnosis was, on average, 3.3 years (range 0.2–45.6 years). There were five patients (6%) with diagnosis before age 18.

The patients were most often (57%) referred to the dermatology clinic from a consultant dermatologist. The duration from obtaining referrals until the first visit at an outpatient or inpatient dermatology clinic was, on average, 32 days. Patients most frequently presented with patches (65%) at the first visit. The most common body areas affected by MF at the onset of the disease were the upper extremities (64%) and lower extremities (70%). The mean Modified Severity-Weighted Assessment Tool (m-SWAT) [31] score at the time of diagnosis was 14.1.

Lab results showed that 10 patients (*n* = 47) had high levels of lactate dehydrogenase (LDH) at debut, and 22 patients (*n* = 61) had high LDH levels at some time during their disease course. Eosinophilia was found in six patients (*n* = 57) at debut and in eight patients (*n* = 68) at some time during their disease.

### 3.3. Staging and Disease Progression

Most of the patients (88%, *n* = 73) were diagnosed with early-stage disease (IA–IIA) (Table 1, Figure 3). Nine patients (10.7%) died, out of whom four (4.8%) died from MF-related complications, such as infection or progressed disease. The death cause of one (1%) patient remains unclear since they were followed at another hospital (Table 1). The overall survival rates were 93% (95% CI: 87–100%) at 5 years, 87% (95% CI: 78–98%) at 10 years, and 77% (95% CI: 63–92%) at 15 years (Figure 4 and Figure 5).

Overall disease progression was seen in 12.5% of the patients. Progression occurred in six men and two women, and the median age was 70 years (range 46–92 years).

Three patients progressed to stage IV: two from stage IB and one from stage IIB. CD30-positive large-cell transformation was diagnosed in three MF patients. The remaining patients had classic MF, apart from one patient who had folliculotropic MF. Stage IA was most associated with remission. In total, 19.6% (*n* = 11) of the patients in this stage showed no evidence of disease activity at their last doctor’s appointment.

### 3.4. Histopathological Findings

The median numbers of histological analyses were 3 (1–11) per patient and 2 (1–6) for MF diagnosis (*n* = 84). When looking at the overall results, the first biopsies demonstrated 51% epidermotropism and 51% atypical lymphocytes (*n* = 83). Less than half (40%) of these patients received a cutaneous lymphoma diagnosis at first biopsy.

The first biopsies of patients with adnexal MF (*n =* 13) revealed that only 23% of patients had suggestive diagnoses of lymphoma, while MF was confirmed by the second biopsy in 69% of cases. An overview of the histopathological variables is presented in Table 3.

### 3.5. Immunohistochemical Findings

The immunophenotypes CD3+/CD4+ were seen in 83/88% of patients (*n =* 78). The antigen CD8+ was found in 65%, and CD8− was found in 16%. All patients debuting in the advanced stage (*n =* 10) exhibited the classic immunophenotypes CD3+/CD4+, and seven of these also exhibited CD30+.

The loss of the CD7 antigen was seen in all patients who underwent immunophenotype testing (*n =* 78), except for nine patients: seven in the early stage and two in the advanced stage.

The antigen CD2 was only found in five patients in the advanced stage (IIB–IVB) and in eight patients in the early stage. An overview of the immunohistochemical variables is presented in Table 3.

### 3.6. T-Cell Receptor Clonality Findings

In total, 75 patients underwent an analysis of TCR clonality with PCR. The majority (*n* = 63) were monoclonal, and the rest (*n* = 7) were polyclonal. The monoclonal receptors showed 6% TCR-beta (TCR-β), 19% TCR-gamma (TCR-γ), 71% had both TCR-γ and β gene rearrangement, and 27% did not show any rearrangement. The adnexal subtypes of MF (*n* = 13) all revealed both monoclonal TCR-γ and β gene rearrangements.

### 3.7. Treatment Modalities

During the study period 26 different treatments were observed among all MF patients: eleven SDTs and eleven systemic therapies (*n =* 84). SDTs were divided into topical therapy, phototherapy, and radiation therapy. Systemic treatments were divided into retinoids, immunotherapy, monoclonal antibodies, and chemotherapy. An overview of these treatments is presented in Figure 6. Topical corticosteroids were the most used treatment and were provided in 97.6% of the patients. They were frequently used in those with early- and advanced-stage disease.

Topical tacrolimus was less used, accounting for 16.7% of the patients with early-stage disease and 3.6% of the patients with advanced-stage disease. Tazarotene and Imiquimod were the least used topical treatments.

The second most used SDTs were ultraviolet B (UVB) and psoralen plus ultraviolet A (PUVA). UVB and PUVA were used in 45% and 39% of the early-stage group, respectively, while UVB was somewhat more used (8%) than PUVA (7%) in patients with advanced-stage disease. Photodynamic therapy (PDT) was the least used phototherapy (2.4%) in the early-stage group, followed by ultraviolet A 1 (UVA 1) phototherapy (12%) in both groups.

Radiation therapy (RT) was used to a greater extent (31%) than Grenz ray therapy (Bucky rays, 5%) in both groups. Total skin electron beam therapy (TSEB) was used in one patient who progressed from the early stage to stage IVB advanced disease. Only 3% of the patients were treated with surgical excision.

Acitretin and methotrexate (MTX) were the most used systemic therapies (21% and 17%) among those in the early-disease group and in those with advanced disease (10% and 6%, respectively).

Chemotherapy was more frequently used in patients with advanced disease. However, in patients with early-stage disease who progressed to an advanced stage, cyclophosphamide, doxorubicin hydrochloride, vincristine sulfate, and prednisone (CHOP) were used more frequently. Only 2% of patients were treated with the monoclonal antibody brentuximab vedotin in both groups. None of the patients received treatment with extracorporeal photopheresis (ECP) or allogenic stem cell transplantation.

#### 3.7.1. Initial Treatment Outcome and Overall Response

Early-stage disease at diagnosis

Patients with early-stage disease most frequently achieved complete response (CR) from PUVA. In total, 17 patients, well over half of the patients, had a complete clearance of skin lesions. For UVB, 10.8% of the patients experienced CR. The highest percentage of patients achieving CR (83%) was observed among those undergoing RT.

Regarding the overall response rate (ORR), the highest rates were seen in PUVA 27/32 (84%) and UVB 17/30 (57%). High rates were also seen in UVA1 (82%), RT (83%), acitretin (47%), Grenz rays (50%), isotretinoin (100%), alitretinoin (100%), and brentuximab vedotin (100%), although very few patients underwent these treatments.

Advanced-stage disease at diagnosis

Among patients with advanced-stage disease at the time of diagnosis, CR was most often achieved from RT (42.9%). The highest ORRs were observed in PUVA (100%), acitretin (50%), and RT (71%). High ORRs were also seen in Grenz rays (50%), chlorambucil (100%), and brentuximab vedotin (50%), but similarly to the early-stage group, only a few patients received this treatment.

Specific values for each group and treatment are depicted in Table 4 and Table 5.

The median follow up times were 6 and 7 years. At the last follow up, 17 (21%) patients achieved CR, 44 (54%) patients achieved partial responses (PR), 8 (10%) patients achieved stable disease (SD), 11 (13%) achieved progressive disease (PD), and 2 (2%) patients experienced relapse (RL) [30].

#### 3.7.2. Combination Therapies

During the study period, combinations of therapies were observed. The most common combination of two treatments was topical corticosteroids in addition to other therapies (42.9%). However, it is difficult to assess how many patients had this combination over time. If one patient was treated with more than one therapy, it was most common to prescribe a combination of SDT and systemic therapy. The combination of phototherapy and retinoids was performed in 11 patients and was the most frequently occurring combined therapy regimen. Nine patients received PUVA and acitretin, which made it the most common combination of a phototherapy and a retinoid. Three patients underwent PUVA in combination with either alitretinoin, isotretinoin, or bexarotene. Two patients had IFN-α and RT in addition to PUVA therapy. Very few patients received other combinations of phototherapy and retinoids (UVB was used in one patient with bexarotene and one with acitretin; alitretinoin was used together with UVA1 in one patient). Treatment with Grenz rays was used in one patient together with chlorambucil and in one with IFN-α. Bexarotene was used in combination with IFN-α in one patient.

## 4. Discussion

### 4.1. General Results

In this study, data from 84 patients with MF in West Sweden were analyzed. The duration from obtaining referrals until the first visit at an outpatient or inpatient dermatology clinic was, on average, 43.6 days, indicating that a reduction in this time should be considered.

### 4.2. Demographic Factors

We found that all patients had other diseases than MF. The association between MF and concomitant diseases had not been extensively studied before. However, it is not surprising that this patient group suffers from age-related disorders since patients with MF are often diagnosed at an older age. It is valuable to consider comorbidity when choosing the most suitable treatment for every patient since it might affect how well a treatment is tolerated.

The patients had various professions, including work in preschools and schools, store assistance, work on a farm, truck driving, office work, construction sites, firefighting, and in electromechanics. Judah et al. suggested that there is a possible link between exposure to environmental or infectious elements and the development of CTCL [32]. There is, however, little research conducted in this area, especially regarding the environmental aspects and the development of CTCL. According to Pahani et al., it is known that exposure to environmental toxicants is a predisposing factor to developing cancer, in general, by exposure to carcinogenic substances [33].

Studies of PCLs in Sweden are scarce, and only some research has been conducted looking at the epidemiological characteristics, clinical outcomes, and treatments of MF in Scandinavia.

In 2009, a Norwegian study observed the incidence of primary CTCL and reported mean ages of 64.4 and 67.3 years for men and women, respectively [4]. A group from Sweden published a study of 44 patients diagnosed with MF and found a median age of diagnosis of 64 years [26]. Recently, another Swedish study reported the median age of 67 years for both MF and SS patients (97 patients) and other CTCLs [28]. However, only adult patients and other types of CTCLs were included in those studies [26,27,28]. Our results showed a median age of 55 years, which was probably explained by the population range at our center, where even children were included, and the fact that in this study we only analyzed patients with MF and not with SS, who are usually older. A retrospective study of 1502 patients in the UK, conducted by Agar et al., looked at the survival outcomes and prognostic factors and found a median age of 54 and a range of 10–89 years [34], similar to our results. Our male-to-female ratio of 1.8:1 was in concordance with all above-mentioned studies, suggesting that men are affected by the disease more often than women.

From the onset of skin symptoms until the initial diagnosis, we observed a median time of 3.3 years (range 0.2–45.6 years) and a mean time of 6.9 years. Studies that have looked at the time from debut to diagnosis have reported a median time of 2 years [35] and a mean time of 4.4 years [36], which are both lower compared to our findings, which in turn is lower than the results from another Swedish study with a median time of 4 years [28]. The reported debuts of skin symptoms according to the medical records are highly subjective, as most patients make an uncertain estimation of when their symptoms appeared. Some patients may suffer from benign inflammatory skin dermatoses not correlating to MF and report them as the MF onset. The results also reflect the prolonged time it takes for the diagnosis of skin lymphomas.

The majority of our patients (88%) had an indolent disease and an indolent course for years before 12.5% of the patients progressed into advanced disease. Studies from the UK and Italy presented similar findings, with 71% and 88% of patients having an early disease stage at diagnosis [34,35].

We found that patients with advanced-stage disease were significantly older (median 66.4 years) at the time of diagnosis than those with early-stage disease (median 50.6 years). A retrospective Austrian study that included 86 adult patients with different variants of PCL found no differences in the age distributions between the early and advanced stages [37]. Advanced-stage disease as well as high age were shown to be independent risk factors associated with poorer overall survival and progressive disease in CTCL, but whether older patients are more likely to be diagnosed with advanced-stage disease seems to be an unexplored topic.

As described before, it is believed that MF lesions appear on the so-called “bathing suiting” areas, most often on the trunk, buttocks, and groin [14,38]. Our study found that most lesions appeared on the upper and lower extremities, but not exclusively, as patients could present with multiple lesions in different body regions.

There was no statistical significance between lesions appearing in sun-exposed or non-sun-exposed body areas on the trunk and upper extremities, but differences were seen in the lower extremities (*p* = 0.028), suggesting that these lesions can appear anywhere on the skin and that uncertain erythematous patches and plaques should be evaluated further.

### 4.3. Immunohistochemical Findings

The loss of antigens has, for a long time, been associated with disease progression. This is especially true for the antigens CD2, CD5, and CD7.

The most reported loss of an antigen from CD4+ T-cells is CD7, which is in line with our results. In a study by Florell et al. [39], this phenomenon was seen as the nonspecific and partial deletion of CD7, which was seen in the majority of both benign and atypical diagnoses. This finding could explain our results of a few patients still having CD7+ antigens on T lymphocytes.

In the same study, it was found that the partial deletion of CD2 is significantly associated with a diagnosis of MF, surely supporting our results of patients in the early stages that received MF diagnosis and did not express the CD2 antigen. However, it is surprising that the patients who still expressed the CD2 antigen were all in the advanced stage, despite it being associated with the progression of the disease [14,40]. The antigen CD2 was only found in five patients in the advanced stage and in eight patients in the early stage in the present study.

### 4.4. T-Cell Receptor Clonality Findings

A Chinese study found that TCR clonality analysis is a useful tool when distinguishing between chronic dermatitis and early-stage MF. All of their patients with MF or suspected MF revealed TCR clonality, while the control group with chronic dermatitis did not [41]. They found that most of their patients had TCR-γ clonality, different from our results, which displayed that most patients had TCR-γ and β clonality. The likelihood of finding clonality is correlated with disease progression and the type of skin lesion [19,42], which could explain the variation in the results. It has indeed been found that the PCR technique can determine a high percentage of monoclonality in the plaque stage but not in the patch stage [42]. It is worth discussing that patients who received clinically and histopathologically verified MF, even in the advanced stages, still expressed polyclonal TCR. This reflects the importance of the clinical evaluation of patients as well as the use of other tools for diagnosing MF.

### 4.5. Treatment Regimens

We found that nearly 97.6% of the patients had used topical corticosteroids at some point, which emphasizes their importance in the treatment of MF, both as a single therapy and as an addition to more potent treatments. Sahlgrenska University Hospital has a long tradition of treating patients with early-stage MF with PUVA, and the results from an internal study performed at the clinic in 2013 showed that 68% of the MF patients received PUVA, while just over 40% were treated with UVB. This study implies that the rate of patients treated with UVB actually increased, as we found that 53.6% of the patients had undergone UVB, but PUVA treatment decreased to 48.8%. The risk of skin cancer, especially nonmelanoma skin cancer, increases with the cumulative dosage of PUVA. Therefore, it is favorable that more patients in the present study underwent treatment with UVB, and hopefully this trend will be maintained. PUVA has long been considered the gold standard for the treatment of early-stage MF, but comparative studies as well as a large review have reported UVB to be a good alternative in terms of the treatment response [43,44]. Though PUVA remains the most effective phototherapy and was associated with the highest ORR in our study (82%), the treatment responses from UVB are also approaching high rates (57%) for ORR in the early stage. To replace PUVA with UVB as a first-line choice could bring several benefits. Not only is it a preventive measure for reducing the incidence of PUVA-related skin cancer, but it is also beneficial for patients with lighter skin types who are more prone to suffer from PUVA-related side effects [44]. However, PUVA is a well-used treatment, and an almost equal number of patients have undergone both treatments.

### 4.6. Treatment Response and Overall Response Rates

When evaluating the treatment response, the calculations were affected by the relatively small numbers of patients receiving each treatment, thus making the sample sizes small. Therefore, we chose to mainly focus on the more frequently used treatments in the following section.

The ORR of 47% for topical corticosteroids in early-stage MF was remarkably low compared to the results from Zackheim et al., who reported an ORR of 90% to class IV topical corticosteroids, predominately in patch-stage MF [45].

The same phenomenon, with lower ORR, is seen in the early-stage group regarding phototherapies: UVB and PUVA. There were generally lower rates regarding CR in both options compared to what was previously reported in early-stage disease.

A previous study reported an ORR of 96% in early-stage MF patients undergoing treatment with narrow-band UVB (NBUVB) [46]. We found an ORR of 53%, where 13.3% of the patients in the early-stage group were estimated to have achieved CR, whereas Gökdemir et al. reported CR in 85.7% of the patients [46]. Thus, our rate of CR was considerably lower. Patients who experienced side effects from the treatment and may have cancelled it prematurely were included, and this may have affected our results. Regarding PUVA, Herrmann et al. found that 79% of patients with stage IA disease achieved CR, as did 59% of patients with stage IB disease, and the ORR for patients with early-stage disease was 95% [47]. We found a slightly lower ORR of 88%. Concerning the ORR of PUVA in patients with advanced-stage disease, we found an ORR of 100%, which was the same rate reported by Hermann et al. [47]. However, with a small sample size, caution must be taken, as these findings might not be applicable on a larger scale.

Acitretin was the most used option of the systemic therapies in advanced-stage patients, with an ORR of 50%. Though acitretin is not a first-line therapy in early-stage disease, 15 patients in this group received acitretin as well. The ORR for all patients was 48%, which resembled the results reported by Cheeley et al., who found an ORR of 59% [47]. However, they reported better outcomes in early-stage patients compared to patients with advanced-stage disease, which differed from our findings, where the results were almost the same, at 47% vs. 50%. One important difference between these studies is the definitions of PR. Cheeley et al. used two definitions: PR1, which was defined as in this study, and PR2 which was defined as an “unequivocal improvement but not meeting PR1”, and this may offer some explanation for the difference in the results [47].

The study describes ORR rates in both groups for any treatment. This finding indicates that treatment in patients with early- and advanced-stage disease, respectively, were accurately adjusted to the disease stage. Consecutively, these results support the current guidelines from the EORTC as well as the Swedish treatment protocol [25,48].

## 5. Conclusions

This study is an important contribution to the knowledge on the epidemiological and clinical features, histopathology, immunophenotype, and molecular findings in patients with diagnosed MF and/or followed up for a long period of time in West Sweden.

## Figures and Tables

**Figure 1 cancers-14-04661-f001:**
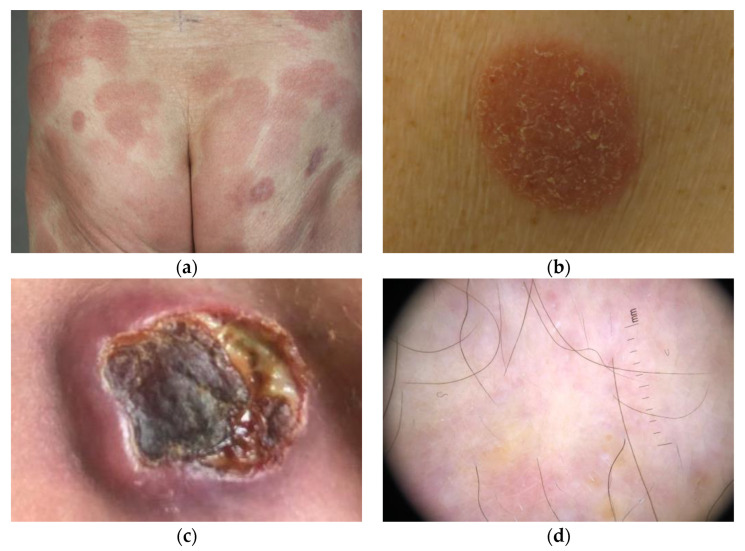
Skin manifestations of mycosis fungoides photographed in some of the study participants. (**a**) Erythematous patches, (**b**) plaque, (**c**) tumor with ulceration, and (**d**) dermoscopy image with orange-yellowish patches.

**Figure 2 cancers-14-04661-f002:**
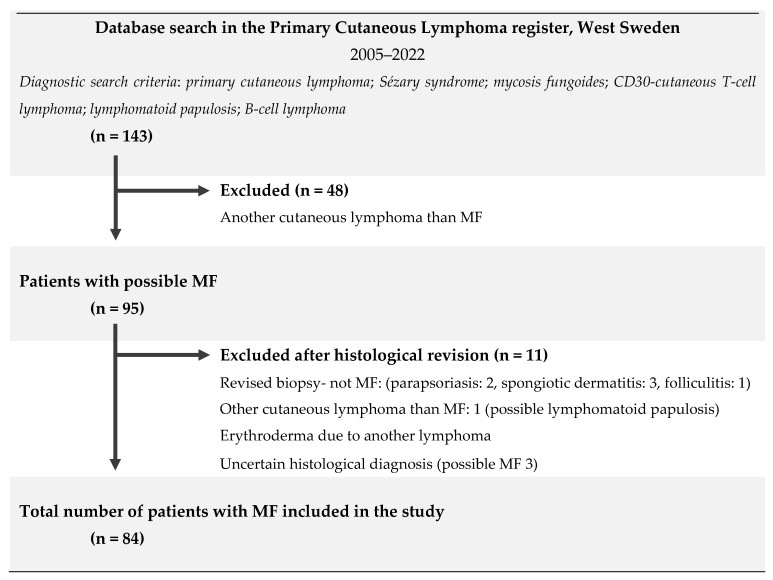
Flowchart of the applied inclusion and exclusion criteria. CD: cluster of differentiation; n: number; MF: mycosis fungoides.

**Figure 3 cancers-14-04661-f003:**
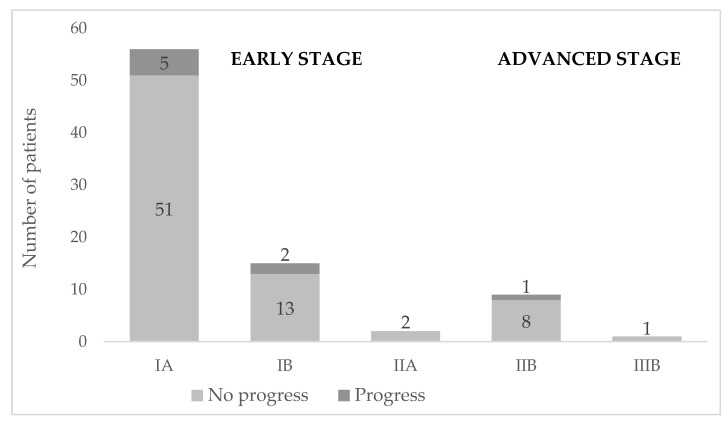
Staging at initial diagnosis of mycosis fungoides and progression (dark grey top of the bars) to more advanced stages (where stages IIB and IIIB are considered advanced-stage disease) during the study period.

**Figure 4 cancers-14-04661-f004:**
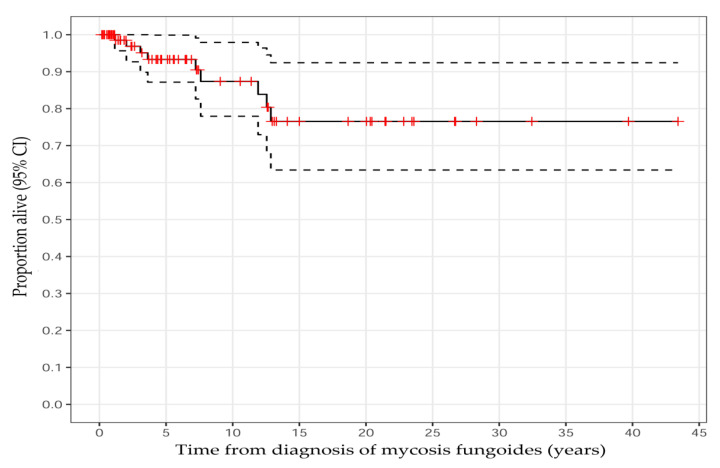
Kaplan–Meier survival plot for all patients with mycosis fungoides. The dotted lines are 95% confidence intervals (CI). The small vertical lines in red denote patients lost to follow-up.

**Figure 5 cancers-14-04661-f005:**
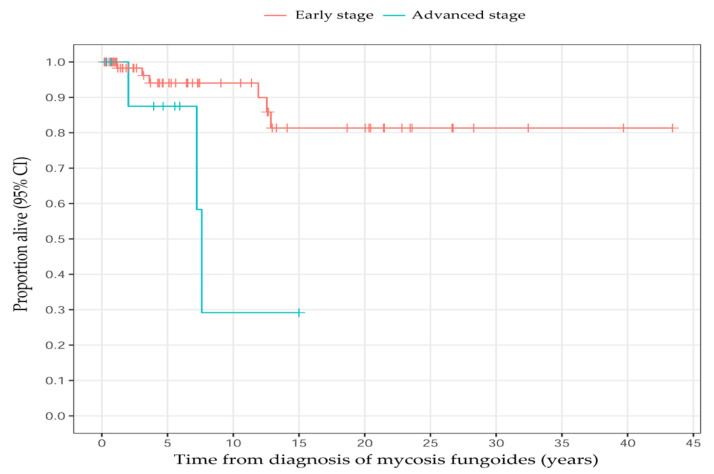
Kaplan–Meier survival plot stratified into early-stage (red) and advanced-stage (blue) disease at the debut of mycosis fungoides. The small vertical lines denote patients lost to follow-up. The estimated hazard ratio with respect to advanced-stage/early-stage disease was 5.3 (95% confidence intervals, CI: 1.3–22; *p* = 0.022).

**Figure 6 cancers-14-04661-f006:**
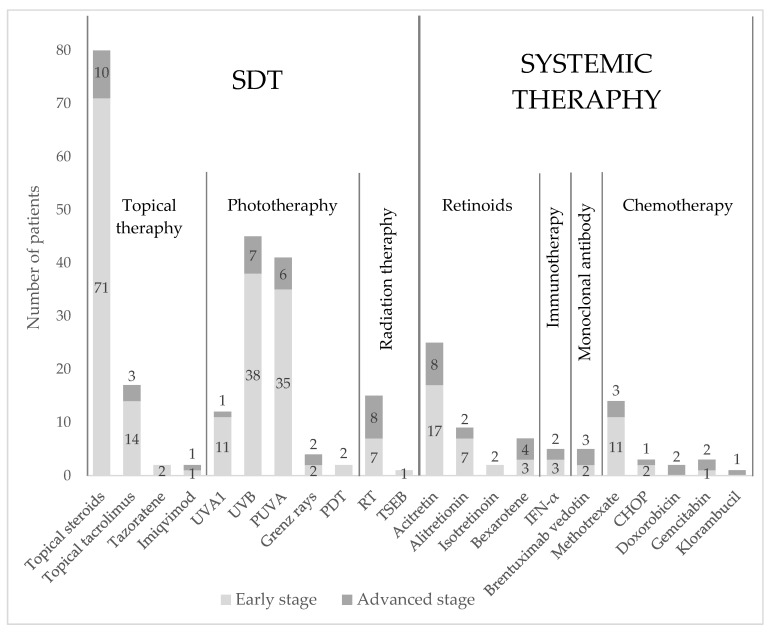
Overview of all treatments, with the digits in the bars presenting the number of patients treated for early-stage (light grey) and advanced disease (dark grey top). SDT: skin-directed therapy; UVA1: ultraviolet A 1; UVB: ultraviolet B; PUVA: psoralen plus ultraviolet A; PDT: photodynamic therapy; RT: radiation therapy; TSEB: total skin electron beam therapy; IFN-α: interferon-alpha; CHOP: cyclophosphamide, doxorubicin hydrochloride, vincristine sulfate, and prednisone.

**Table 1 cancers-14-04661-t001:** Summary of demographic and clinical characteristics of 84 patients with mycosis fungoides.

Characteristics	Values
Age at time of diagnosis, years, median (range)	55.4 (8.6–91.8)
Mean ± SD	52.4 ± 20.5
Sex (male/female), n (%)	
Male	54 (64.3)
Female	29 (33.5)
Time from onset of skin symptoms to initial diagnosis, years, median (range)	3.3 (0.2–45.6)
Time from referral to first visit at the clinic, days, median (range)	32 (0–338)
Diagnosis received after first visit at dermatology clinic, n (%)	
Yes	61 (73.5)
No	22 (26.5)
No data	1 (1.2)
Clinical features, debut, n (%)	
Papule	19 (22.6)
Macule	15(17.9)
Patch	54 (64.3)
Plaque	33 (39.2)
Nodule	1 (1.2)
Tumor	7 (8.3)
Poikiloderma	1 (1.2)
Hypopigmentation	5 (5.9)
Hyperpigmentation	5 (5.9)
Erythroderma	3 (3.6)
Body area of skin disease, debut, n (%)	
Head	16 (19)
Face	12 (14.3)
Scalp	4 (4.8)
Trunk	41 (48.8)
Sun-exposed	26 (31)
Unexposed to the sun	21 (25)
Upper extremity	53 (63)
Sun-exposed	28 (33)
Unexposed to the sun	32 (33)
Lower extremity	58 (69)
Sun-exposed	26 (31)
Unexposed to the sun	41 (48.8)
Percentage of skin area involved at time of diagnosis, n (%)	
<10%	63 (75)
≥10%	21 (25)
m-SWAT at time of diagnosis, median (range)	4 (0–156)
Mean ± SD	14.1 ± 29.8
Clinical stage at time of diagnosis, n (%)	
IA	56 (67.5)
IB	15 (18.1)
IIA	2 (2.4)
IIB	9 (10.8)
IIIA	0
IIIB	1 (1.2)
IVA1	0
IVA2	0
IVB	0
No data	1 (1.2)
Staging, n (%)	
Early (IA–IIA)	73 (88)
Advanced (IIB–IVB)	10 (12)
Types, n (%)	
Classic MF	63 (75)
Folliculotropic MF	9 (10.7)
Syringotropic MF	4 (4.8)
Hypopigmented MF	1 (1.2)
CD30-positive transformation	7 (8.3)
Skin type according to Fitzpatrick, n (%)	
I	3 (3.75)
II	55 (68.8)
III	20 (25)
IV	2 (2.5)
No data	4 (5)
Remission at last appointment, n (%)	14 (16.7)
Deceased patients, n (%)	
MF-related death	4 (4.8)
Other causes of death	4 (4.8)
No data	1 (1.2)

SD: standard deviation; n: number; MF: mycosis fungoides; CD: cluster of differentiation; m-SWAT: Modified Severity-Weighted Assessment Tool.

**Table 2 cancers-14-04661-t002:** Summary of malignant comorbidities in 25 patients with mycosis fungoides.

Skin Cancers	Total
Basal cell carcinoma	14
Squamous cell carcinoma in situ	4
Squamous cell carcinoma	3
Melanoma in situ	2
Lentigo maligna	1
Lentigo malignant melanoma	1
Malignant melanoma	2
Merkel cell carcinoma	1
Hematological malignancies	**Total**
Anaplastic large-cell lymphoma	1
Chronic lymphocytic leukemia	1
Chronic myelomonocytic leukemia	1
Diffuse large B-cell lymphoma	1
Epstein–Barr virus–positive diffuse large B-cell lymphoma)	2
Follicular lymphoma	1
Hodgkin lymphoma	1
Marginal zone lymphoma	1
Solid tumors	**Total**
Prostate cancer	4
Breast cancer	3
Esophagus cancer	1
Endometrial cancer	1
Urinary bladder cancer	1
Colon cancer	1
Rectal cancer	1
Myxofibrosarcoma	1

**Table 3 cancers-14-04661-t003:** Summary of histopathological and immunohistochemical variables of 84 patients with mycosis fungoides.

Characteristics	Values
Histopathological characteristics, n (%)	
Epidermotropism	
Yes	55 (69.6)
No	19 (24)
uncertain	3 (3.8)
Atypical lymphocytes	
Yes	67 (84.8)
No	9 (11.4)
uncertain	2 (2.5)
TCR clonality, n (%)	
Monoclonal	63 (75)
Gamma	60 (71.4)
Beta	49 (58)
Immunohistochemistry, n (%)	
CD2+	13 (16.7)
CD3+	65 (83.3)
CD4+	68 (87.2)
CD5+	20 (25.6)
CD7+	9 (11.5)
CD8−	13 (16.7)
CD8+	51 (65.4)
CD20−	6 (7.7)
CD20+	24 (30.8)
CD30−	14 (17.9)
CD30+	26 (33.)
CD45+	4 (5.1)

n: number; TCR: T-cell receptor; CD: cluster of differentiation.

**Table 4 cancers-14-04661-t004:** Initial treatment outcomes for different skin-directed therapy modalities in patients with early-stage disease and advanced-stage disease at the time of MF diagnosis.

SDT Treatments	CR	PR	SD	PD	ORR		NA	Patients
Topical therapy								
Topical steroids	6	26	29	11	32	(44%)	12	72
Early stage	6	23	25	7	29	(47%)	12	62
Advanced stage	-	3	3	4	3	(30%)	-	10
Topical tacrolimus	-	4	7	1	4	(33%)	5	12
Early stage	-	4	5	1	4	(40%)	4	10
Advanced stage	-	-	2	-	-		1	2
Tazarotene	-	-	2	-	-		-	2
Early stage	-	-	2	-	-		-	2
Advanced stage	-	-	-	-	-		-	-
Imiquimod	-	1	1	-	1	(50%)	-	2
Early stage	-	1	-	-	1	(100%)	-	1
Advanced stage	-	-	1	-	-		-	1
Phototherapy								
UVA1	2	7	2	-	9	(75%)	1	12
Early stage	2	7	1	-	9	(82%)	1	11
Advanced stage	-	-	1	-	-	-	-	1
UVB	4	15	10	7	19	(51%)	8	37
Early stage	4	13	10	4	17	(57%)	7	30
Advanced stage	-	3	-	3	3	(43%)	1	7
PUVA	17	17	2	2	34	(89%)	3	38
Early stage	17	10	2	2	27	(84%)	3	32
Advanced stage	-	6	-	-	6	(100%)	-	6
Grenz rays	1	1	1	1	2	(50%)	-	4
Early stage	1	-	-	1	1	(50%)	-	2
Advanced stage	-	1	1	-	1	(50%)	-	2
PDT	-	1	1	-	1	(50%)	-	2
Early stage	-	1	1	-	1	(50%)	-	2
Advanced stage	-	-	-	-	-		-	-
Radiation therapy								
RT	7	3	3	-	10		1	13
Early stage	4	1	1	-	5	(83%)	1	6
Advanced stage	3	2	2	-	5	(71%)	-	7
TSEB	-	1	-	-	1	(100%)	-	1
Early stage	-	1	-	-	1	(100%)	-	1
Advanced stage	-	-	-	-	-		-	-

SDT: skin-directed therapy; CR: complete response; PR: partial response; SD: stable disease; PD: progressive disease; ORR: overall response rate; NA: not applicable; UVA1: ultraviolet A 1; UVB: ultraviolet B; PUVA: psoralen plus ultraviolet A; PDT: photodynamic therapy; RT: radiation therapy; TSEB: total skin electron beam therapy.

**Table 5 cancers-14-04661-t005:** Initial treatment outcomes for different systemic treatment modalities in those with early-stage disease and advanced-stage disease at the time of diagnosis.

Systemic Treatments	CR	PR	SD	PD	ORR		NA	Patients
Retinoids								
Acitretin	1	10	6	6	11	(48%)	3	23
Early stage	1	6	5	3	7	(47%)	3	15
Advanced stage	-	4	1	3	4	(50%)	-	8
Alitretinoin	-	2	-	1	2	(67%)	6	3
Early stage	-	1	-	-	1	(100%)	6	1
Advanced stage	-	1	-	1	1	(50%)	-	2
Isotretinoin	-	2	-	-	2	(100%)	-	2
Early stage	-	2	-	-	2	(100%)	-	2
Advanced stage	-	-	-	-	-	-	-	-
Bexarotene	-	-	1	3	-	-	3	4
Early stage	-	-	-	3	-	-	-	3
Advanced stage	-	-	1	-	-	-	3	1
Immunotherapy								
IFN-α	-	-	1	3	-	-	1	4
Early stage	-	-	-	2	-	-	1	2
Advanced stage	-	-	1	1	-	-	-	2
Monoclonal antibody								
Brentuximab vedotin	-	2	1	-	2	(67%)	1	3
Early stage	-	1	-	-	1	(100%)	1	1
Advanced stage	-	1	1	-	1	(50%)	-	2
Chemotherapy								
Methotrexate	-	3	6	4	3	(23%)	1	13
Early stage	-	2	5	2	2	(20%)	1	10
Advanced stage	-	-	2	2	-	-	-	3
Doxorubicin	-	-	1	1	-	-	-	2
Early stage	-	-	-	-	-	-	-	-
Advanced stage	-	-	1	1	-	-	-	2
Gemcitabin	-	-	1	2	-	-	-	3
Early stage	-	-	1	-	-	-	-	1
Advanced stage	-	-	-	2	-	-	-	2
Chlorambucil	-	1	-	-	1	(100%)	-	1
Early stage	-	-	-	-	-	-	-	-
Advanced stage	-	1	-	-	1	(100%)	-	1

CR: complete response; PR: partial response; SD: stable disease; PD: progressive disease; ORR: overall response rate; NA: not applicable; IFN-α: interferon-alpha; CHOP: cyclophosphamide, doxorubicin hydrochloride, vincristine sulfate, and prednisone.

## Data Availability

The data presented in this study are available upon reasonable request to the corresponding author.

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
