# Peer review of "Diagnostic Outcomes and Treatment Modalities in Patients with Mycosis Fungoides in West Sweden—A Retrospective Register-Based Study"

_cancers, 2022, doi:10.3390/cancers14194661_

Round 1

Reviewer 1 Report

Abstract

Page 1 – Line 24 “…. Advanced WILL BE evaluated”

Page 3 –  Line 90 “Excluded after revision (n=15) Uncertain histological and/or clinical diagnosis” which one? Described the clinical and histopathological criteria used to establish the MF diagnosis. Use pictures to describe this.

Page 4 – Line 144 – The description described fourteen other malignancies with six other lymphomas. But 27 had skin cancer such as BBC. “Fourteen (16.7%) patients had other malignancies, six (7%) other lymphomas, and 27 (32%) skin cancer such as BCC (15.5%). Other benign skin conditions than 145 MF occurred in 40 patients (47.6%).” The description is confusing and skin cancer was considered no cancer. A specific description of synchronous or asynchronous malign tumors and a table with a specific diagnosis is better to describe the clinically associated lesion as “other malign tumor”

Page 7 – The authors described 26 different treatments and 88 patients were diagnosed with early-stage disease (IA-IIA). The research bias does not permit the comparative analysis. So many treatments and so many early-stage patients. The treatment could be classified into  Skin-directed therapy, Systemic therapy, Immunotherapy, Monoclonal antibody as targeted therapy, and Radiation therapy. other similar general descriptions could be applied.

            Apply Kaplan Meier or actuarial table to identify survival, progression rates, and various clinical and histopathological variables.

Discussion

            The 26 different treatments do not permit eliminating beta error.

Page 13 – Lines 408-419 – Was immunohistochemistry used to auxiliary the MF diagnosis?

Page 13 – Line 422 – The high use of topical corticosteroids does not indicate its use. A better survival rate indicates that.

Page 14 – Lines 476-479 – The 26 different treatments do not permit eliminating beta error. It is not possible to compare initial and advanced MF.

Page 14 – Lines 481-482 – The study does not describe histopathological findings in patients with MF.

Author Response

Cancers (ISSN 2072-6694)

AUTHORS COMMENTS TO  REVIEWER

Manuscript ID cancers: 1838084
Wojewoda et al

Stage 1. Diagnostic outcomes and treatments modalities in patients with Mycosis Fungoides in West Sweden- a retrospective register-based study

Sincerely,

Karolina Wojewoda/authors

Reviewer 2 Report

Mycosis fungoides (MF) is a rare cutaneous lymphoma with chronic course, and it has a heterogenous exhibitions, especially at its early stage. Dermatologists and researchers have been studying hard for MF’s epidemiology, clinical features, diagnosis, and treatments. However, due to the rarity and indolence of MF, findings are often limited by sample sizes and time periods. Wojewoda et al here reports a retrospective register-based study in a 17-year period for 84 MF patients in West Sweden. The detailed clinical and laboratory information from this report will help dermatologists and oncologists for their clinical practice.  

Overall, the manuscript is well written, data (tables/Figure) are properly presented. Except double checking all numbers or frequencies (%)/words/spaces in the manuscript, I have no further comments.

Author Response

Cancers (ISSN 2072-6694)

AUTHORS COMMENTS TO THE EDITOR AND REVIEWER

Manuscript ID cancers: 1838084
Wojewoda et al

Stage 1. Diagnostic outcomes and treatments modalities in patients with Mycosis Fungoides in West Sweden- a retrospective register-based study

Thank you for your letter and giving us the opportunity to submit a revised draft of the manuscript ’’ Diagnostic outcomes and treatments modalities in patients with Mycosis Fungoides in West Sweden- a retrospective register-based study ’’

We appreciate the time and effort that you dedicated to providing feedback on our manuscript and are grateful for the insightful comments, which contributed to valuable improvements on our paper. We have incorporated most of the suggestions made by the editor and reviewers. Those changes are marked in red colour.

We hope that the revised manuscript will better suit the Cancers  but are also pleased to consider further revisions, and we thank you for your continued interest in our research.

Sincerely,

Karolina Wojewoda/authors

Reviewer #2

(x) English language and style are fine/minor spell check required 

Response: Spell checking has been done accordingly, thank you.

Mycosis fungoides (MF) is a rare cutaneous lymphoma with chronic course, and it has a heterogenous exhibitions, especially at its early stage. Dermatologists and researchers have been studying hard for MF’s epidemiology, clinical features, diagnosis, and treatments. However, due to the rarity and indolence of MF, findings are often limited by sample sizes and time periods. Wojewoda et al here reports a retrospective register-based study in a 17-year period for 84 MF patients in West Sweden. The detailed clinical and laboratory information from this report will help dermatologists and oncologists for their clinical practice.  

Overall, the manuscript is well written, data (tables/Figure) are properly presented. Except double checking all numbers or frequencies (%)/words/spaces in the manuscript, I have no further comments.

Response: Thank you for this excellent observation. We checked and changed all numbers or frequencies (%)/words/spaces in the manuscript.

Round 2

Reviewer 1 Report

None